# TwinDNN: A Tale of Two Deep Neural Networks

## Abstract

Compression technologies for deep neural networks (DNNs), such as weight quantization, have been widely investigated to reduce the DNN model size so that they can be implemented on hardware with strict resource restrictions. However, one major downside of model compression is accuracy degradation. To deal with this problem effectively, we propose a new compressed network inference scheme, with a high accuracy but slower DNN coupled with its highly compressed DNN version that typically delivers much faster inference speed but with a lower accuracy. During inference, we determine the confidence of the prediction of the compressed DNN, and infer the original DNN for the inputs that are considered not confident by the compressed DNN. The proposed design can deliver overall accuracy close to the high accuracy model, but with the latency closer to the compressed DNN. We demonstrate our design on two image classification tasks: CIFAR-10 and ImageNet. Our experiments show that our design can recover up to 94% of accuracy drop caused by extreme network compression, with more than 90% increase in throughput compared to just using the original DNN. This is the first work that considers using a highly compressed DNN along with the original DNN in parallel to improve latency significantly while effectively maintaining the original model accuracy.

## 1 Introduction

Machine learning is one of the most popular fields in the current era. It is used in various ways, such as speech recognition, face recognition, medical diagnosis, etc. However, the problem is that the neural networks for machine learning applications Krizhevsky et al. (2012); He et al. (2016a;b) are becoming too large and slow to be on a small chip for real-time systems. As a result, there has been a significant amount of research to reduce the size of the neural networks so that their inference latencies are low enough to handle real-time inputs Zhang et al. (2020); Zhou et al. (2016); Zhang et al. (2018). There are quite a few approaches to compress existing neural networks, but for field-programmable gate arrays (FPGAs), quantization of network is the most popular and effective method to reduce the size and inference latency at the same time Han et al. (2016). In particular, extremely low bit-width networks on FPGAs, such as binary or ternary neural networks have been studied recently Wang et al. (2018); Courbariaux & Bengio (2016); Courbariaux et al. (2015); Zhao et al. (2017); Yao Chen & Chen (2019); Li & Liu (2016); Blott et al. (2018). These networks provide an extra benefit that normal quantized networks do not provide in terms of multiplier (DSP) utilization. The idea is that extremely low bit-width weights allow multiplications to be done in conditional logic, which can be implemented by logic gates, without using a special hardware for multiplication (DSP). This fact can allow developers to utilize additional DSPs in other ways where DSPs can be useful. However, this benefit is not free, of course. One major downside of these low bit-width networks is that they tend to have even more accuracy drop than regular quantized neural networks, as a result of further reduced precision. Therefore, it is more difficult to use binary or ternary neural networks as they are, especially in the fields such as surveillance or medical diagnosis systems, where the cost of that accuracy drop is much larger than the latency improvement.

The goal of this study is to accelerate neural network inferences by using an extremely low bit-width network implementations on FPGAs, while maintaining the accuracy of the original network by using relatively high precision network concurrently, without having to develop a single DNN accelerator that meets both accuracy and latency requirements. The main contribution is to find a

mechanism to choose the right network to infer for specific inputs, and this is done by creating a hierarchical structure of two different compressed networks and utilizing the output of the initial inference to determine the need of extra verification.

In summary, we propose a system that consists of two distinct networks: one extremely low bit-width network that is focused on latency, and the other moderately quantized network that is focused on accuracy. In this paper, extremely low bit-width network will be called a compressed network, and moderately quantized network will be called an original network. These two networks work in a way that can exploit advantages in both latency and accuracy at the same time. The overall mechanism is similar to the one presented in Mocerino & Calimera (2014). However, there has not been any study of this concept in FPGA accelerators for deep neural networks. This represents a novel direction in neural network research, to pair compressed network and original network in parallel in order to improve latency while maintaining the original accuracy. Our main contributions are as follows:

- Accelerators that are designed and optimized to exploit low bit-width networks, with pipelined and parallelized computation engines that use a minimum number of DSPs as possible.

- A software solution that allows two accelerators to be run in hierarchical fashion, utilizing confidence of a compressed network prediction.

- For ImageNet and ResNet-18, our TwinDNN solution can deliver up to $1.9\times$ latency improvement with only 3% of extra DSPs used for compressed network, and up to 95% of accuracy loss is recovered during hierarchical inference.

In Section 2, some background information related to this work will be introduced. In Section 3, design flow of our implementation and experiment will be explained. In Section 4, the results of our experiments will be described. Section 5 will conclude the paper with future explorations.

## 2 BACKGROUND

### 2.1 EXTREMELY LOW BIT-WIDTH NEURAL NETWORKS

Recent researches have succeeded in binarizing or ternarizing parts of layers in neural networks Wang et al. (2018); Courbariaux & Bengio (2016); Courbariaux et al. (2015); Zhao et al. (2017); Yao Chen & Chen (2019). Many experiments claim that these compression methods are very effective in terms of latency reduction with some accuracy drops. As one would expect, as the number of bits used to represent either weights or feature maps decreases, accuracy drops more significantly. Because the goal of our study is to compensate for the accuracy loss caused by compression, we can forgive moderate accuracy loss, as long as the benefit of using those networks is significant. The following equations:

$$
\begin{aligned}
w_b &= \begin{cases} -w_{scale} & \text{if } b = 0 \\ +w_{scale} & \text{if } b = 1 \end{cases} \\
w_t &= \begin{cases} -w_{scale} & \text{if } t = -1 \\ +w_{scale} & \text{if } t = 1 \\ 0 & \text{if } t = 0 \end{cases}
\end{aligned}
\tag{1}
$$

show how these extremely low bit-width weights are used in computation. $b$ is a 1-bit value that can be either 0 or 1, and $t$ is a 2-bit value that can take either -1, 0, or 1. The key idea here is that $w_{scale}$ value is the same across the weights. The bits are only used in sign representations. In binary, as an example, a single bit of 0 represents negative and 1 represents positive, and this logic can be implemented in a simple condition, or a multiplexer in FPGAs. $w_{scale}$ value is stored separately, and the same $w_{scale}$ value is multiplied over all binary weights to get the actual weight values. However, we do not need to perform all of these multiplications separately. Consider $b1 = 0$ and $b2 = 1$ for the binary case. $a$ stands for activation, or feature map, then we can express a very simple neural network computation as follows:

$$
\begin{aligned}
a_{next} &= w_{b1} \times a_1 + w_{b2} \times a_2 \\
&= -w_{scale} \times a_1 + w_{scale} \times a_2 \\
&= w_{scale} \times (-a_1 + a_2)
\end{aligned}
\tag{2}
$$

This shows how binary and ternary weight computations can be handled with a single multiplication. Reducing the number of actual multiplications reduces the need of DSPs, and indeed makes the overall computation faster. For ternary, the only difference is that two bits now represent positive, negative, and zero. Therefore, the main benefit of using extremely low bit-width neural networks is more effective and balanced resource utilization, specifically on FPGAs. For a typical DNN implementation on FPGAs, DSP is the one that directly determines the performance, and so is the limiting factor of the performance. Therefore, typical DNN implementations on FPGAs utilize nearly all the DSPs available, and other resources are left under-utilized. Extremely low bit-width networks, on the other hand, only uses a minimal number of DSPs, and mainly utilizes look-up tables (LUTs), which are generally left unused for computation (e.g., only used for control logic). In this study, we instantiate both typical DNN (original DNN) and extremely low bit-width DNN (compressed DNN) at the same time, in a way that original DNN uses most of the DSPs available on the board, and compressed DNN uses extra LUTs that were not used by original DNN. This method allows us to utilize both DSP and LUT resources as much as possible to ultimately reduce the overall latency.

## 2.2 Final Layer Outputs in Neural Network

In neural network image classifications, output of the final layer is a list of values for each class, and the class with the highest final layer output is typically chosen as a prediction. Here, each value represents how possible is that image in the class, and we will call these values probabilities for convenience. Note that in order to compute the actual probabilities, we need to apply a softmax function to these values. However, we do not apply a softmax function here, because we only need to find the class with the highest value. Anyways, the point is that the output of a neural network provides more information than just a prediction. Specifically, it provides information about all probabilities of predictions that the input can be classified as. It is definitely possible to utilize this additional information to enhance the prediction. In this study, we will make use of the probability of the second most possible label. From compressed network inference output, along with the prediction itself, we also need to find out whether to infer the original network or not. The probability of the second most possible label is used here to determine if the prediction of compressed network is confident enough to be used as an actual output without verification from the original network.

Here is an example of how we can utilize this information during inference. For handwritten digit recognition, let's define $Out(x)$ as final output value for label $x$. There is no problem if $Out(0) = 0.9$ and $Out(1), \ldots, Out(9) < 0.1$, because the network is almost sure that the digit is 0. However, in case where $Out(1) = 0.5$ and $Out(2) = 0.4$, although $Out(1)$ is greater than $Out(2)$, we cannot guarantee that 1 is actually a correct label, because that 0.1 difference could have been resulted from the noise of using a low precision network. This is what we use as a confidence of the prediction. Instead of just finding a label with maximum probability, the system will now find two labels with first and second maximum probabilities, and compute the difference between those two probabilities. If the difference is large (i.e., beyond a threshold determined empirically), it means that the label with highest probability dominates other labels, so compressed network prediction can be considered confident. If the difference is small, it means that two labels with highest probabilities both have potentials to become a true label, so the prediction is considered not confident. In this case, the input needs additional verification from the original network that is designed to have maximum accuracy.

## 3 Implementation

Our implementation flow consists of three parts: creating original network and compressed network models, implementing high-level-synthesis (HLS) accelerator intellectual properties (IPs) for those networks, and creating a software system for hierarchical inference.

## 3.1 MODEL GENERATION

Creating original network starts with a typical floating-point training, which can also be completed by using a pre-trained model available. For training, we used a Caffe framework Jia et al. (2014), which was also customized to be used by other works (e.g., Yao Chen & Chen (2019)). In order to enhance the accuracy, we use a variety of well-known techniques, such as learning rate decay and batch normalization. Specifically, batch normalization of activations allows us to standardize and stabilize the outputs, which can speed up the training process. After these networks are trained to the point where accuracy does not improve anymore, we merge these batch-normalization layers into convolutional layers using

$$
\begin{aligned}
w_{new} &= \frac{\gamma \times w_{old}}{\sqrt{Variance}} \\
b_{new} &= \frac{\gamma \times (b_{old} - Mean)}{\sqrt{Variance}} + \beta
\end{aligned}
\tag{3}
$$

where $Mean, Variance, \gamma$, and $\beta$ are the parameters trained in batch-normalization layers, $w_{old}$ and $b_{old}$ are weight and bias values before merge, and $w_{new}$ and $b_{new}$ are those value after merge. This is typically done in neural networks to reduce the extra latency of normalizing the activations during inference, and it gives us exactly the same inference results.

The network weights are then quantized to designated bit-widths. Quantization scheme is determined by accuracy drop and distribution of weights. First, we try a uniform quantization scheme. We always use uniform quantization whenever possible because non-uniform quantization requires extra logic and computation required for bit shifting in hardware. If accuracy drop is significant, we then try a non-uniform quantization scheme depending on the distribution of weights and activations. There can still exist a slight accuracy drop after non-uniform quantization, and there are few ways presented in Zhao et al. (2019); NVIDIA to recover this accuracy drop, which can be implemented in the future.

Compressed network model, on the other hand, cannot be generated without a training scheme that is specifically designed for binary and ternary neural networks. For binary neural network, which was used in our CIFAR-10 experiment, we used the same model in Zhao et al. (2017), which was trained using the method proposed by Courbariaux & Bengio (2016). This model showed approximately 5% accuracy drop compared to the original network model. As explained in Zhao et al. (2017), there are some advanced training techniques available, but they are not used in our work. Although it is still better to have a higher accuracy for the compressed network, it is not necessary especially for the purpose of this work, as we will show how our design can recover the accuracy drop caused by extreme network compression.

For ternary neural network, which was used in our ImageNet experiments, we trained the model by using the framework explained in Yao Chen & Chen (2019). Our trained model, however, could not reach the exact accuracy reported in Yao Chen & Chen (2019), and this is due to additional fine-tuning and data augmentations that they performed. Our trained model also showed approximately 5%-8% accuracy drop compared to the original network model, which seems valid for the purpose of this work.

## 3.2 ACCELERATOR DEVELOPMENT

Xilinx's Vivado high-level-synthesis tool was used to generate IPs for both original and compress networks. Their tools allow developers to apply various optimizations, such as loop pipelining and array partitioning, more easily on their FPGAs. We targeted Ultra96 and ZCU102 FPGAs, which are both Arm-based Xilinx Zynq UltraScale+ MPSoC development boards. ZCU102 has more overall resources than Ultra96, and is used for MobileNetV2 experiment only.

Figure 1 shows the overall architecture of accelerators. For convolutional and fully connected layer computations, the main technique we used was to have multiple pipelined computation engines that compute partial multiply-accumulate (MAC) operations. It will perform element-wise multiplication of weights and features, and then compute the sum of the products using an adder tree. These computation engines are pipelined so that they can produce a MAC of 16 weights and 16 features

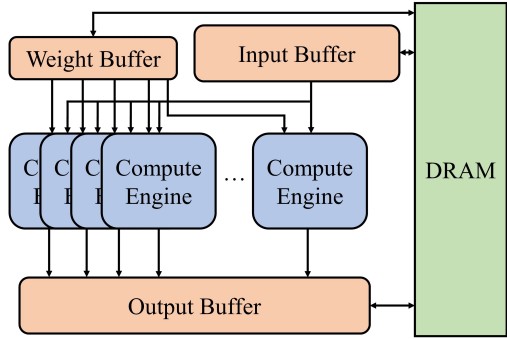

Figure 1: Basic accelerator architecture

every single cycle. For 16-bit network, 16 of these computation engines are instantiated to serve 256 elements in parallel, by using 256 DSPs, and these compute engines are shared between layers for maximum resource utilization. As a further optimization for 8-bit network, we can double the efficiency of using DSPs using the method proposed by Xilinx.

For binary and ternary networks, computation engines do not use any DSPs. Instead, multiplexers (MUX) are used to determine the sign of the weights. For binary networks, $w_b$ is used as a 1-bit selector to determine the output between $-A$ and $+A$. For ternary networks, $w_t$ is used as a 2-bit selector to determine the output between $-A$, 0, and $+A$. Then, the sum of those outputs will be computed using the adder tree, same as before. At the end of all computations, we will multiply $w_{scale}$ values from Equation 1.

### 3.3 SOFTWARE DEVELOPMENT

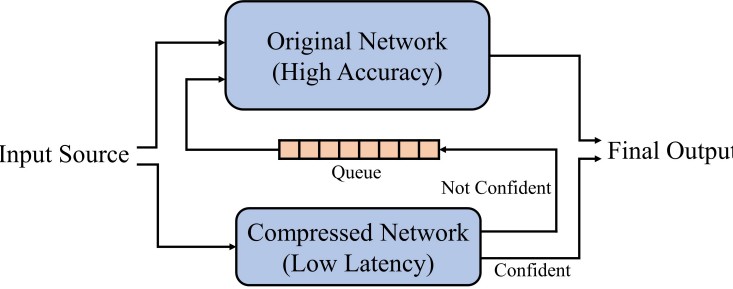

Figure 2: Graphical representation of hierarchical architecture

The accelerators are invoked from the software running inside the processing system of the FPGA. Because two accelerators are both instantiated in a single design, they support concurrent execution. Figure 2 shows a graphical representation of the software system, which is designed to fully utilize both networks. First, an image from the source to be processed will wait for either network to become idle. Whatever network that becomes idle first will perform the initial inference for that image. Note that for the original network, idle means that its queue is empty as well. If the original network was used for the initial inference, its prediction will always be used as the final prediction, because it has a higher accuracy. However, if the compressed network was used for the initial inference, the software will compute the confidence of the compressed network prediction to determine if the image needs additional inference on the original network. Confidence is defined as the difference between two largest output values. If confidence is above the threshold, the prediction of the compressed network will be used as a final prediction. If confidence is below the threshold, however, the input image will go into the queue for the original network inference. Meanwhile, when the image from the source starts initial inference, the next image becomes ready immediately to wait for either network to become idle. This is to ensure that both accelerators are running for the entire time until

Table 1: Resource utilization ratio of accelerators

| Dataset | Network | Precision | DSP | LUT |
|---------|---------|-----------|-----|-----|
| CIFAR-10 | ResNet-18 | 16-bit | 256 | 20110 |
| | ConvNet | Binary | 4 | 25074 |
| | | 16-bit + Binary | 260 | 63727 |
| ImageNet | ResNet-18 | 16-bit | 274 | 24114 |
| | ResNet-18 | 8-bit | 274 | 30970 |
| | ResNet-18 | Ternary | 8 | 25416 |
| | | 16-bit + Ternary | 282 | 56922 |
| | | 8-bit + Ternary | 282 | 63710 |
| | MobileNetV2 | 32-bit | 536 | 60424 |
| | MobileNetV2 | Ternary | 64 | 27507 |
| | | 32-bit + Ternary | 600 | 119851 |

Table 2: Accuracy and latency comparison for all configurations

| Dataset | Network | Precision | T* | Accuracy (%) | R* (%) | Latency (ms) |
|---------|---------|-----------|-----|--------------|--------|--------------|
| CIFAR-10 | ResNet-18 | 16-bit | | 94.1 | | 391 |
| | ConvNet | Binary | | 89.6 ($\triangledown4.5$) | | 64 ($6.1\times$) |
| | | 16-bit + Binary | 1.5 | 92.8 ($\triangledown1.3$) | 71.1 | 77.5 ($5.0\times$) |
| ImageNet | ResNet-18 | 16-bit | | 69.5 | | 306 |
| | ResNet-18 | 8-bit | | 67.9 ($\triangledown1.6$) | | 255 ($1.20\times$) |
| | ResNet-18 | Ternary | | 63.6 ($\triangledown5.9$) | | 244 ($1.25\times$) |
| | | 16-bit + Ternary | 0.7 | 69.2 ($\triangledown0.3$) | 94.9 | 160 ($1.91\times$) |
| | | 8-bit + Ternary | 1.0 | 67.1 ($\triangledown0.8$) | 81.4 | 153 ($1.66\times$) |
| | MobileNetV2 | 32-bit | | 69.8 | | 231 |
| | MobileNetV2 | Ternary | | 62.2 ($\triangledown7.6$) | | 288 ($0.80\times$) |
| | | 32-bit + Ternary | 0.3 | 68.5 ($\triangledown1.3$) | 82.9 | 140 ($1.65\times$) |

T*: Threshold, R*: Accuracy Recovery ($1 - \frac{AccuracyDrop_{combined}}{AccuracyDrop_{compressed}}$ (%))

all the images are processed, and is the most important difference between this work and Mocerino & Calimera (2014). In contrast to the CPU implementation of Mocerino & Calimera (2014), where the worst case latency is combined latency of original and compressed networks, our FPGA parallel implementation allows both networks to run in parallel for the entire time until the input source is depleted, so the amortized worst case latency is just the latency of the original network.

Threshold value is determined from experiment. Threshold value of 0 means all the compressed network predictions will be considered confident, and none of the inputs will go into the queue. This results in both networks running in parallel independently, as original network will also get the input from source. High threshold value means that more images go into the original network, thus it results in high accuracy and high latency. Note that when the threshold value goes above a certain point, the queue will contain some images even after all images from the source are depleted. From that moment, only the original network will be running, and this reduced parallelism impacts the latency significantly. Therefore, it is recommended to choose a threshold value that will keep the queue small. Threshold value of infinity, in fact, is the same as just running the original network alone, because all compressed network output will be considered not confident and require original network inference. We test a variety of threshold values to see which one gives a most balanced result between accuracy and latency, and will use it to obtain the final result.

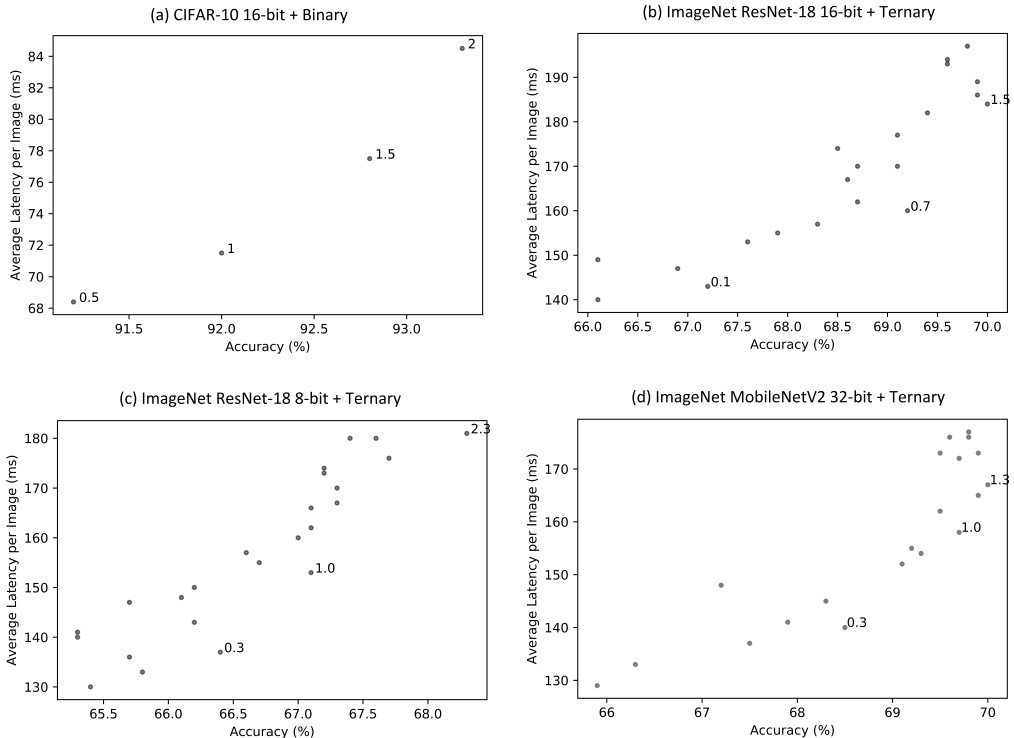

Figure 3: Accuracy and latency plots for different experiment configurations

# 4 EXPERIMENT

## 4.1 CIFAR-10

We first tested our design on CIFAR-10 dataset. The experiment was performed with a 16-bit ResNet-18-based network created by us, and a binary ConvNet-based network created by Zhao et al. (2017), on Ultra96 development board, with a frequency of 100MHz. Table 1 shows the resource utilization ratio for individual accelerators. As expected, 16-bit network mainly uses DSPs. 256 DSPs were used for $16 \times 16$ computation engines. Binary network, on the other hand, only uses 4 DSPs, which are used for $w_{scale}$ multiplication. It uses more LUTs than 16-bit network since it mainly performs computation on LUTs. Table 1 also shows the resource utilization ratio for the entire design. Note that the entire design uses more LUTs than the sum of two accelerators. This is because extra LUTs are used for interconnects and memory interfaces. Although LUT usage for individual models may seem small, we are actually utilizing more than 90% of LUT resources available for the TwinDNN solution.

Figure 3 (a) shows the accuracy and latency plot for different threshold values. We chose 0.5, 1, 1.5, and 2 as our threshold values. Higher threshold value resulted in higher latency and accuracy. A threshold value of 1.5 gives the most balanced result between latency and accuracy, thus we used it for the final experimental result. Table 2 shows the comparison between configurations, with 16-bit network as a baseline. A combination of 16-bit and binary network was able to recover more than 71% of accuracy loss caused by binarizing the network. Average latency is $5\times$ faster than the baseline network, which proves we are using the binary network effectively.

Note that the binary network results in much higher latency than the result reported in Zhao et al. (2017). This is because first, we used much lower frequency than Zhao et al. (2017), and second, we converted their accelerator built with Vivado SDSoC platform into Vivado HLS accelerator with our customized software part that supports parallel inference of two accelerators.

## 4.2 ImageNet

Next, we tested our design on a much bigger dataset, ImageNet. This time the experiment was performed with two different networks on different FPGA configurations: ResNet-18 on Ultra96 with 150MHz frequency and MobileNetV2 on ZCU102 with 200MHz frequency.

We use 16-bit, 8-bit, and ternary versions of ResNet-18, and 32-bit and ternary versions of MobileNetV2. Ternary networks were used as compressed networks, and other networks were used as original networks. Table 1 shows that similar to our CIFAR-10 experiment, ternary network uses much less number of DSPs compared to other fixed-point networks. Also, for ResNet-18, 8-bit network uses more LUTs than 16-bit network, and this is because 8-bit network uses additional logic for bit shifting and introduces additional parallelism from using 1 DSP for 2 multiplications. Note that ternary networks are parallelized to the extent where they would fit on the FPGAs along with original network accelerators, so they do not have much speedup compared to baseline original networks. The meaningful point is that we can combine these two accelerators in parallel to increase throughput with only a small number of extra DSPs compared to original network accelerators.

Figure 3 (b)-(d) shows the accuracy and latency plots for different threshold values. This time, we tested more variety of threshold values so that as much design space is explored as possible. In these graphs, only the points, or threshold values, with the most balanced and efficient results are labeled, for a clear view. For ResNet-18 16-bit and ternary network configuration, threshold value of 0.7 gives the most balanced result between latency and accuracy. For ResNet-18 8-bit and ternary network configuration, threshold value of 1.0 gives the most balanced result, and for MobileNetV2 32-bit and ternary configuration, threshold value of 0.3 gives the most balanced result. After a threshold value of about 1.0, accuracy of combined network becomes almost identical to that of original network. This result suggests that our software can identify the majority of inputs that are likely to be incorrect with compressed network by using confidence.

Note that for some high threshold values, final accuracy is higher than the accuracy of original network. We believe that this is just a result of noise, where there are few cases ternary network predicts correctly while 16-bit network does not. However, there is a chance that these two networks actually complement each other in terms of accuracy. This means that each network is more accurate on certain types of inputs, and our hierarchical design can exploit this hypothesis by finding the confidence of a prediction instead of just a prediction. This can be one of the areas where we can study further.

Table 2 shows the final result and comparison with original networks as baselines. Note that our 8-bit network does not show the ideal speedup of $2\times$, expected from DSP usage, and this is because there are other factors, such as memory bandwidth, that also limit the speed of the inference. For ResNet-18 16-bit and ternary configuration, with a threshold value of 0.7, our design was able to recover almost 95% of accuracy loss caused by network compression, with more than $1.91\times$ reduced latency compared to 16-bit network alone. For 8-bit and ternary configuration, with a threshold value of 1.0, our design was able to recover more than 81% of accuracy loss, with $1.66\times$ reduced latency compared to 8-bit network alone. Note that 16-bit and ternary configuration gives better results than 8-bit and ternary configuration overall. This is because our 8-bit model is just a low-precision version of 16-bit model. If we perform additional retraining for 8-bit model, 8-bit and ternary configuration is expected to have a better result than what is currently reported. Finally, for MobileNetV2 32-bit and ternary configuration, with a threshold value of 0.3, our design was able to recover 82.9% of accuracy loss, with $1.65\times$ reduced latency.

Overall result indicates that our design works well for accuracy recovery given two optimized neural network accelerators, with only a small amount of extra inferences. Our definition of confidence is also proved to be a useful metric of verifying neural network output.

## 5 Conclusion

In this paper, we proposed a TwinDNN system with a high-accuracy network and a low-latency network using a hierarchical inference logic that will infer high-accuracy network when the prediction of low-latency network is not considered confident. This design becomes especially more effective on the FPGAs where DSP resources are limited compared to LUT resources, as compressed network

latency will solely depend on the number of LUTs. There are several aspects that make this study stand out. The first aspect is its high flexibility. Although in this project we mostly used ResNet-18, we can basically put any two networks on the same dataset. There are already many neural network accelerators that are built for different focuses: accuracy and latency. We only need to find two accelerators that would fit on the target FPGA, and simply apply the same logic for experiment. Use of Zhao et al. (2017) shows this idea, although we did not reach their exact performance. Second is better concentration. Accelerator development becomes much more difficult when developers need to care about multiple aspects at the same time. However, this work can potentially allow one group of developers to solely focus on increasing accuracy, and the other group of developers to solely focus on reducing latency. It will ultimately reduce the time and effort it takes to build a high-quality accelerator that achieves both accuracy and latency goals.

There are also several aspects where this work can be enhanced further. The first aspect is specialized training. If we can train a specialized network that is trained only to classify between top few predictions of the compressed network, we may be able to save resources and improve the confidence of the compressed network. Another specialized training scenario can be to train the compressed network to classify or detect easy objects and train the original networks to target difficult objects. This way, the two networks can complement each other better. Second is heterogeneous computing with graphics processing unit (GPU). GPUs are usually much more efficient than FPGAs on floating-point operations, and floating-point precision indeed gives higher accuracy compared to low bit-width fixed-point precision. If we can make GPU run the original network as floating-point, and make FPGA run the compressed network, we may be able to achieve an even more efficient solution for this study.

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
