# OpenReview forum: "TwinDNN: A Tale of Two Deep Neural Networks"
_ICLR.cc/2021/Conference — Reject_

### Official Review · AnonReviewer1 · 2020-10-26
**Cascade CNN implemented on FPGAs**

**Rating:** 4
**Confidence:** 4

**Review:**

SUMMARY:
The paper at hand discusses a compressed network inference scheme, where two networks are trained to solve a give classification task. One network aims at achieving a high accuracy, whereas the other network is a highly compressed network which is able to highly speed-up inference. The compression of the network is performed by quantizing the parameters of the network to two or to three different values. The approach of the authors consists of estimating the uncertainty of the highly-compressed network by thresholding the scores. In the case that one score does not exceed the other scores by the pre-defined threshold, the sample will additionally be feed into the high-accuracy network. Experiments are performed on the CIFAR-10 and the ImageNet dataset using a convolutional neural network, a ResNet and a MobileNetV2 network. The speed-up is evaluated with respect to specialized hardware (FPGAs).

REASONS FOR SCORE:
1. The novelty of the described approach is very limited. The approach of training two different neural networks, one for high accuracy and one for low latency, was already proposed by Mocerino & Calimera (2014). Also estimating the uncertainty of the prediction by analyzing the score vectors should be considered common-knowledge.
2. The description of batch normalization in Section 3.1 is wrong. From the description it seems like the weights and biases are normalized, which is not the case for batch-normalization where activations are normalized!
3. The mathematical writing is flawed at various places, e.g., Equation (1) is not properly integrated in the sentence. The same holds for Equation (2).
4. Overall, the paper does not explain the specialized hardware used to implement the proposed approach. There are many abbreviations like HLS, IPS, FPGA, DPS and LUT that are not even defined. This makes the paper hard to understand for readers that are not familiar with FPGAs. In my opinion the paper needs a rewrite before publication.
5. Missing labels for points in Figure 3 b)-d). It is not clear to the reader what the unlabeled grey dots show. Perhaps, the interpretabilty of this plots could be improved by adding color and a colorbar?
6. Are the numbers in Table 2 test accuracies? If yes, how are the thresholds determined? This should have been done using a validation set.

DECISION:
Overall, I recommend rejecting this paper because of the issues stated above.

UPDATE AFTER REBUTTAL:
I would like to thank the authors for their responses. After reading the updated version I still would recommend to reject the paper. The reason is that the paper is written for a very narrow audience and is hard to understand for readers who are not familiar with this area of research. Also I  feel like some of my concerns where not properly addressed in the updated version (e.g., issue 2 and 6). After reading the authors' response, I still think that the novelty of this paper is very limited. I decided to keep my score at 4.

MINOR REMARKS AND TYPOS:
- The last sentence of introduction: section numbers are roman instead of arabic as in the template and "section" should be capitalized.

---

> ### Author Response · Authors · 2020-11-18
> **Response to AnonReviewer1**
>
> Thanks for your time and effort spent on reviewing this paper.
>
> Q: The novelty of the described approach is very limited.
>
> A: I agree that the idea of having two different neural networks for inference in a hierarchical fashion has already been used is previous works. However, there are some differences and new contributions in this work that I would like to point out. Please refer to the general comment that I made for further discussion.
>
> Q: The description of batch normalization in Section 3.1 is wrong. From the description it seems like the weights and biases are normalized, which is not the case for batch-normalization where activations are normalized!
>
> A: Here, what I wanted to say was that we could modify the original weights and biases in order to remove batch-normalization layer, where activations are normalized. However, I understand my description of batch-normalization may be misleading. I will rewrite that description for a clearer explanation of what batch-normalization layers actually do.
>
> Q: The mathematical writing is flawed at various places, e.g., Equation (1) is not properly integrated in the sentence. The same holds for Equation (2).
>
> A: I will make appropriate changes to my equations so that they are properly integrated in the sentence.
>
> Q: Overall, the paper does not explain the specialized hardware used to implement the proposed approach. There are many abbreviations like HLS, IPS, FPGA, DPS and LUT that are not even defined. This makes the paper hard to understand for readers that are not familiar with FPGAs. In my opinion the paper needs a rewrite before publication.
>
> A: I agree that it would be better to give some explanations on the abbreviated terms that I use. I will make changes accordingly.
>
> Q: Missing labels for points in Figure 3 b)-d). It is not clear to the reader what the unlabeled grey dots show. Perhaps, the interpretabilty of this plots could be improved by adding color and a colorbar?
>
> A: I have intentionally omitted labels for the most of dots on graphs, and have only included labels for the points that provide the most balanced results. This is because I thought including labels for all the points may make this graph hard to see, and only one of the points is actually used as a final result. I will add some more explanations to describe what unlabeled dots represent.
>
> Q: Are the numbers in Table 2 test accuracies? If yes, how are the thresholds determined? This should have been done using a validation set.
>
> A: The numbers in Table 2 are test accuracies. I was not thinking my experiment as a threshold selection process, but rather a process of trying all possible threshold values. However, in order to choose a threshold value as one of the parameters for my system, it is correct that threshold choosing dataset should be separated from the real test dataset.
>
> Q: The last sentence of introduction: section numbers are roman instead of arabic as in the template and "section" should be capitalized.
>
> A: I will make appropriate changes for that.

---

### Official Review · AnonReviewer3 · 2020-10-28
**Cascade CNN idea (use simple network when possible, if not confident, use larger more accurate network)**

**Rating:** 4
**Confidence:** 4

**Review:**

The paper claims that this is the first work to consider "using a high compressed DNN along with the original DNN in parallel". The idea is to use the compressed DNN when possible and to fall back on the original DNN when the confidence in the prediction is low.

This is an interesting idea and area, as the authors outline there is considerable scope to improve throughput/latency (or save power) with such an approach. The authors report they can reduce latency by nearly 2x at a low cost. Experiments are performed using an FPGA. The compressed networks are created using binary or ternary neural networks.

Unfortunately, I believe this is an idea that has been explored by numerous previous works (and in some cases evaluated using FPGAs) e.g.: CascadeCNN ( https://arxiv.org/abs/1807.05053 )

Of course, cascade architectures, e.g. for keyword spotting are also common. The idea of anytime predictors and early exit may also be relevant here and even schemes such as dynamic channel pruning (https://arxiv.org/abs/1810.05331). Interesting work has also been undertaken to provide improved confidence metrics for such approaches.

Some related work:
Big/little DNN: https://ieeexplore.ieee.org/document/7331375
https://arxiv.org/abs/1710.03368
BranchyNet: https://arxiv.org/abs/1709.01686
https://arxiv.org/abs/1708.06832
etc.

It is unclear what specific new contribution the paper makes over previous work?

Would there be an argument to compare to a pruned network?

It would be useful to indicate the size of each network too (rather than just DSP/LUT counts), again to help comparisons to previous work.

UPDATE AFTER REBUTTAL: Many thanks to the authors for their comments. I still believe novelty is limited and will leave my score unchanged.

---

> ### Author Response · Authors · 2020-11-18
> **Response to AnonReviewer3**
>
> Thanks for your time and effort spent on reviewing this paper.
>
> Q: It is unclear what specific new contribution the paper makes over previous work?
>
> A: I agree that the idea of having two different neural networks for inference in a hierarchical fashion has already been used is previous works. However, there are some differences and new contributions in this work that I would like to point out. Please refer to the general comment that I made for further discussion.
>
> Q: Would there be an argument to compare to a pruned network?
>
> A: Currently, optimization of neural network on FPGAs is mainly focused on quantization, and this is because quantization is the most straightforward and efficient method that makes FPGA accelerators smaller and faster. Pruning, on the other hand, is not well studied yet for FPGA implementations, due to its nature of randomness, and therefore, it would be difficult to implement an efficient FPGA accelerator that can fully exploit the benefits of pruning.
>
> Q: It would be useful to indicate the size of each network too (rather than just DSP/LUT counts), again to help comparisons to previous work.
>
> A: Although it is also an important measurement of neural networks, size of the network is not a concern in this work. It is because networks are first stored on SD card, which is much larger than popular networks in current era. It is true that the network will not fit on on-chip memory. In order to solve this, during inference, we constantly load partial weights into a weight buffer, as shown in Figure 1, so that you only need a constant size of weight buffer for any size of network. However, I am open to add any extra comparisons to previous work, in order to distinguish my work from previous work. Would size of network still be a good choice on that?

---

### Official Review · AnonReviewer2 · 2020-10-30
**This paper presents a method to balance the precision and latency of Deep Neural Networks, which is good but but not good enough.**

**Rating:** 5
**Confidence:** 3

**Review:**

## writing
-pros
	- The language is concise and easy to understand, and can clearly convey their own academic views
	- The overall quality of the article is nice, and the structure of the paper is very complete. The engineering ability reflected in this article is also relatively strong.
- cons
	- Some of the points that may be questioned are not explained. e.g. Why on the latencies of ResNet and mobile-net, the model with the highest compression rate is slower than the one proposed in this article, when structure is chose as ResNet-18.
## originality
- pros
	- Innovation can be guaranteed
-cons
	- The validity of confidence as the output of network has been concerned in the previous research or application.
## significance
- pros

	- This paper presents a method to balance the precision and latency of Deep Neural Networks. The core idea of this method is to take the output confidence of the compressed network as the criterion. If the confidence levels of different classes differ greatly (i.e. exceeding the threshold determined by experience), it means that the label with the highest probability dominates other tags, so the prediction of compressed network can be considered as reliable. Otherwise they are likely to be unreliable. In this case, the input requires additional validation from the original network. For the model that can be input into FPGA, this method proposes an idea to accelerate the parallel operation of highly compressed network and original uncompressed network. This method enables developers and researchers to determine the balance point between speed and precision by adjusting the trust degree of the output confidence of compressed network (by setting threshold). It used to be a discrete choice, but now it's continuously adjustable. In addition, such a network also makes the quantization compression technology more simple and of more application value.
- cons
 - The core idea of compressed network, on the one hand, is to obtain higher real-time performance, on the other hand, because the volume of large-scale network is too large. Therefore, this method is equivalent to sacrificing one party to the other. And we cannot use this method with a network unless the uncompressed version of the network is able to be stored in the FPGA, which greatly limits the application.
-The theoretical explanation of confidence is not enough, and there are no corresponding more controlled experiments, so the persuasiveness is not strong enough. Because this article should be based on experiments, but the number of experiments is still relatively small, which may need to be further supplemented, especially more about the comparison of low-precision quantization networks such as 2bit and 3bit with the methods mentioned in the article. The exchange of precision and speed is equivalent to solving the transition problem before the quantization network and the actual network. But is it possible to get the same result by directly quantifying it with a slightly higher number of digits, which is better in storage cumsuming?

---

> ### Author Response · Authors · 2020-11-18
> **Response to AnonReviewer2**
>
> Thanks for your time and effort spent on reviewing this paper.
>
> Q: Why on the latencies of ResNet and mobile-net, the model with the highest compression rate is slower than the one proposed in this article, when structure is chose as ResNet-18.
>
> A: I am assuming your question is on performance of ternary implementations. Ternary implementations do not show that great performance (sometimes even worse than original network), because of the resource limitations. The performance of accelerators is directly related to the resources that they use. Resource usage of compressed networks is limited to the resources that are left unused in original network, and sometimes, the amount of unused resources is so small, that its ternary network shows worse performance than the original network.
>
> Q: The validity of confidence as the output of network has been concerned in the previous research or application.
>
> A: I understand that the concept of "confidence" has already been used in previous research. However, there are some differences and new contributions in this work that I would like to point out. Please refer to the general comment that I made for further discussion.
>
> Q: The core idea of compressed network, on the one hand, is to obtain higher real-time performance, on the other hand, because the volume of large-scale network is too large. Therefore, this method is equivalent to sacrificing one party to the other.
>
> A: The third paragraph of section 1 describes that we call a moderately compressed network an "original network". Therefore, original networks are not actually full sized networks and are typically small enough to fit on FPGAs, although it really depends on the size of the target FPGA.
>
> Q: And we cannot use this method with a network unless the uncompressed version of the network is able to be stored in the FPGA, which greatly limits the application
>
> A: Generally, the size of the network itself is not an issue in FPGA implementation, because networks are typically stored in SD card. However, it is true that the network will not fit on on-chip memory. In order to solve this, during inference, we constantly load partial weights into a weight buffer, as shown in Figure 1, so that you only need a constant size of weight buffer for any size of network.
>
> Q: But is it possible to get the same result by directly quantifying it with a slightly higher number of digits, which is better in storage cumsuming?
>
> A: As described in previous two questions, storage consumption is not a main concern. Also, it is difficult to implement a network with slightly higher number of digits (say 4-bit) for this experiment. It is because the main benefit of using binary/ternary networks is that they do not use multipliers (DSPs). However, for a 4-bit network, its implementation without a multiplier would be quite inefficient, as it will use a massive amount of LUTs for 16:1 muxes.
>
> However, I do agree that it would be better to have additional experiments. The main issue is the training time of a ternary version of large neural networks. For the rebuttal, I might be able to add few smaller neural networks to my experiment.

---

### Official Review · AnonReviewer4 · 2020-11-03
**Good work but improvements are needed**

**Rating:** 4
**Confidence:** 4

**Review:**

Summary:

This paper proposes a framework to accelerate DNN inference on small embedding systems using an extremely low bit network and a moderately quantized network jointly. The mechanism of the proposed work is to first compute the difference using top2 prediction scores from the compressed network to determine if the inference is further needed for the original network. The proposed framework is evaluated on both CIFAR10 and ImageNet with different network structures and the empirical results indicate better accuracy and latency over baselines.


Pros:
- The idea of how to utilize both compressed and original networks is novel.
- The framework has high flexibility that the network component can be replaced by any other networks based on user demands.
- The experiments show some promising results on both accuracy and latency.


Cons:
-  The authors state that w_scale value is the same across the weights in equation (1). However, it is not clear to see how it is implemented.
- In section 2.2, I am confused about the statements “...output of the final layer is …., and this is very similar to the probability…””. Is softmax included? Why not use the outputs with softmax?
- Many terms are not well-explained in the paper, such as FPGA, IPs (page 4), LUT(table 1). I recommend the authors to give some explanations on these terms for the audience without experience in this domain.
- Figure 2 seems to be confusing. Why are there two inputs to the original network if the input of the original network is determined by the compressed network?
- In table 2, “accuracy recovery” is not clearly defined.
- Caption for Figure 3 is missing, e.g., configurations. In addition, I recommend making the makers bigger.
- Overall, the presentation of the paper needs to be improved and further clarifications are needed for some parts.

---

> ### Author Response · Authors · 2020-11-18
> **Response to AnonReviewer4**
>
> Thanks for your time and effort spent on reviewing this paper.
>
> Q: The authors state that w_scale value is the same across the weights in equation (1). However, it is not clear to see how it is implemented.
>
> A: For this answer, I will only talk about ternary networks, because both binary and ternary networks require a similar logic. Section 2.1 basically describes how ternary neural networks are implemented [1]. The main feature of and ternary neural network is that most of the weight values are ternary (-1, 0, 1) value times some scalar value ($w_{scale}$). These ternary weight values, including $w_{scale}$, are determined during quantization-aware training scheme in [1]. A very simple neural network computation is as follows: $a' = a_1w_1 + a_2w_2$, where a denotes input feature, w denotes weight, and a' denotes output feature. This computation requires two multiplies. However, let's say that w_1 = $w_{scale}$ and w_2 = -$w_{scale}$. Then, we can convert the original equation into $a' = (a_1 – a_2)w_{scale}$, where it now requires one multiplication. This is where the benefit of ternary neural network comes from. If this explanation is what you needed to clearly understand how it is implemented, I can add this to section 2.1.
>
> Q: In section 2.2, I am confused about the statements “...output of the final layer is …., and this is very similar to the probability…””. Is softmax included? Why not use the outputs with softmax?
>
> A: Output of the final layer in this work means the output before the softmax layer. In fact, we do not use Softmax at all layer in this work, because we do not need a probability distribution, but only need argmax and the difference between top values. However, I understand that my statements may have caused confusions. I will make changes to those statements to make it clearer.
>
> Q: Many terms are not well-explained in the paper, such as FPGA, IPs (page 4), LUT(table 1). I recommend the authors to give some explanations on these terms for the audience without experience in this domain.
>
> A: I agree that it would be better to give some explanations on the abbreviated terms that I use. I will make changes accordingly.
>
> Q: Figure 2 seems to be confusing. Why are there two inputs to the original network if the input of the original network is determined by the compressed network?
>
> A: Original network also takes inputs from the input source, because in case queue is empty, we do not want to let original network to be idle. This process is explained in section 3.3, and was implemented in order to maximize the throughput and distinguish this work from CPU implementations.
>
> Q: In table 2, “accuracy recovery” is not clearly defined.
>
> A: I will add a simple explanation of this term
>
> Q: Caption for Figure 3 is missing, e.g., configurations. In addition, I recommend making the makers bigger.
>
> A: The captions (configurations) are written above each graph in Figure 3. I can make the texts on graphs larger for a clearer view.
>
> [1] Y. Chen et al. T-DLA: An Open-source Deep Learning Accelerator for Ternarized DNN Models on Embedded FPGA. ISVLSI, 2019.

---

### Author Response · Authors · 2020-11-18
**General Comments from Authors**

Thank you all for your time and effort spent on reviewing this paper. All of you gave thorough and insightful reviews, which could definitely help further improve my work.

I have noticed that few reviewers questioned the novelty of this work, and I would like to provide some general comments to answer this. I agree that the idea of having two different neural networks for inference in a hierarchical fashion has already been used is previous works [1] [2]. However, there are some differences that I would like to point out.

First, the goal of previous works is generally focused only on latency and accuracy, but in this work, we have an additional goal, which is resource usage. For a general neural network implementation on FPGA, number of DSPs (multipliers) is a limiting factor, which means there are some leftover LUTs (look-up tables). In this study, we are only utilizing these leftover LUTs with a very limited number of DSPs, and this is why we only use binary/ternary neural networks for compressed networks. In contrast, most previous works use relatively high bit-width neural networks (4 bit for [1]) for compressed networks, which uses decent amount of DSPs, and this may could sacrifice the performance of the original network by a lot.

Second, this work uses a relatively recent method of training compressed networks, as proposed in [3]. Most previous works did not do any retraining and quantization-aware training for compressed networks, so their overall accuracies were relatively low. For example, in [1], the smallest bit width it could achieve was 4-bit, and the network became unusable beyond that point, because of massive accuracy drop. However, in this work, even binary and ternary networks have reasonable accuracies, which allow us to exploit additional benefit described in the previous paragraph.

Finally, this work uses a true real-time parallel inference scheme, which implements both neural networks on a single design. For previous works, original networks are only used when compressed network results are not considered confident, and this may result in original networks to become idle if all compressed network results are considered confident. Furthermore, [1] does not use any parallel architecture, but instead, does an FPGA reconfiguration for original network inference after compressed network inferences are done, which may not be able to provide real time inference results in some cases. However, as described in section 3.3 of this paper, our parallel inference scheme forces the original network to be busy for the entire time, which gives higher throughput and more stability especially for the worst-case scenario.

[1] A. Kouris et al. CascadeCNN: Pushing the Performance Limits of Quantisation in Convolutional Neural Networks. FPL, 2018.

[2] L. Mocerino et al. CoopNet: Cooperative Convolutional Neural Network for Low-Power MCUs. CoRR, 2014.

[3] Y. Chen et al. T-DLA: An Open-source Deep Learning Accelerator for Ternarized DNN Models on Embedded FPGA. ISVLSI, 2019.

---

### Decision · Program_Chairs · 2021-01-07
**Final Decision**

**Decision:**

Reject

**Comment:**

All four knowledgeable referees have indicated reject mainly because the novelty is limited - they thought (and I also agreed) that  it would be difficult to argue the novelty of the proposed framework simply by considering the more recent compressed network training technique, as the reviewer mentioned through rebuttal. In addition, there were concerns about various terms and basics specialized for hardware that are not kindly explained for more diverse audiences in the machine learning field. It improved a little through revision, but I think it needs a more kind explanation. It seems that more thorough experimental verifications  are needed.